# Calcium Signaling Regulates Autophagy and Apoptosis

**DOI:** 10.3390/cells10082125

**Published:** 2021-08-18

**Authors:** Pramod Sukumaran, Viviane Nascimento Da Conceicao, Yuyang Sun, Naseem Ahamad, Luis R Saraiva, Senthil Selvaraj, Brij B Singh

**Affiliations:** 1Institute for Health Promotion Research, Department of Population Health Sciences, UT Health San Antonio, San Antonio, TX 78229, USA; sukumaran@uthscsa.edu; 2Department of Periodontics, School of Dentistry, University of Texas Health Science Center, 7703 Floyd Curl Drive, Mail Code 7763, San Antonio, TX 78229, USA; Nascimentoda@uthscsa.edu (V.N.D.C.); suny8@uthscsa.edu (Y.S.); naseem.ahamad.uthscsa@gmail.com (N.A.); 3Research Department, Sidra Medicine, Doha P.O. Box 26999, Qatar; saraivalmr@gmail.com (L.R.S.); senthil.bio@gmail.com (S.S.)

**Keywords:** calcium signaling, calcium channels, autophagy, apoptosis, stem cells, neuronal and immune cell function

## Abstract

Calcium (Ca^2+^) functions as a second messenger that is critical in regulating fundamental physiological functions such as cell growth/development, cell survival, neuronal development and/or the maintenance of cellular functions. The coordination among various proteins/pumps/Ca^2+^ channels and Ca^2+^ storage in various organelles is critical in maintaining cytosolic Ca^2+^ levels that provide the spatial resolution needed for cellular homeostasis. An important regulatory aspect of Ca^2+^ homeostasis is a store operated Ca^2+^ entry (SOCE) mechanism that is activated by the depletion of Ca^2+^ from internal ER stores and has gained much attention for influencing functions in both excitable and non-excitable cells. Ca^2+^ has been shown to regulate opposing functions such as autophagy, that promote cell survival; on the other hand, Ca^2+^ also regulates programmed cell death processes such as apoptosis. The functional significance of the TRP/Orai channels has been elaborately studied; however, information on how they can modulate opposing functions and modulate function in excitable and non-excitable cells is limited. Importantly, perturbations in SOCE have been implicated in a spectrum of pathological neurodegenerative conditions. The critical role of autophagy machinery in the pathogenesis of neurodegenerative diseases such as Alzheimer’s, Parkinson’s, and Huntington’s diseases, would presumably unveil avenues for plausible therapeutic interventions for these diseases. We thus review the role of SOCE-regulated Ca^2+^ signaling in modulating these diverse functions in stem cell, immune regulation and neuromodulation.

## 1. Introduction

Calcium is a prominent regulator for diverse processes such as gene transcription, proliferation, cell motility, cell signaling, neuronal regulation, autophagy and apoptosis [1]. To perform such a broad spectrum of functions, the cells have evolved multiple unique mechanisms that are modulated by different proteins that regulate cellular Ca^2+^ levels. In addition, the spatial and temporal regulation of Ca^2+^ levels that is maintained by various Ca^2+^ channels, transporters, pumps, and their binding to key proteins, play an essential role in maintaining a tight control on intracellular Ca^2+^ levels. The transient receptor potential (TRP) channels have been studied extensively and play a prominent role in regulating various cellular functions [2,3,4]. The TRPC subfamily consists of seven members (TRPC1-7) with diverse modes of regulation and physiological function. Intracellular Ca^2+^ plays a crucial role in both basal and induced autophagy [5,6]. Plenty of evidence has suggested a complex role of Ca^2+^ in the regulation of autophagy as well as in the regulation of apoptosis [7]. However, the mechanism by which Ca^2+^ controls autophagy and apoptosis remains controversial [8]. Previous studies have shown a negative role of Ca^2+^ on regulating autophagy that induces apoptosis [5,9], while many studies showed a positive role of Ca^2+^ in activating autophagy [1,8,10,11]. Nonetheless, Ca^2+^-permeable channels have emerged as important factors in modulating both basal and induced autophagy that can also prevent apoptosis. In mammalian cells, starvation causes autophagosomes to form omegasomes at the endoplasmic reticulum (ER). Autophagy is induced either by mammalian target of rapamycin (mTOR) inactivation, or adenosine monophosphate (AMP) activated protein kinase (AMPK) activation, which causes distinct Unc-51-like autophagy activating kinase 1 (ULK1) activation [12,13,14] (Figure 1A). Altered Ca^2+^ transmission has been implicated in a variety of processes fundamental for both non-excitable and excitable cells. Here, we review what is known about Ca^2+^-channel-induced Ca^2+^ signaling and how this fundamental second messenger regulates life and death decisions, with particular attention directed to cell autophagy in both excitable and non-excitable cells.

## 2. Autophagy and Apoptosis Crosstalk Is Guided by Ca^2+^ Influx

Intracellular Ca^2+^ plays a crucial role in both basal and induced autophagy along with modulating apoptosis [5,6] (Figure 1A,B). Autophagy, or “self-eating”, is a highly conserved and dynamic catabolic process termed as a fundamental cell survival system, intracellular clearance pathway and major intracellular degradation mechanism. Cellular stress, including damaged mitochondria, protein aggregation, bacterial and viral infections, tumors, hypoxia, aging and metabolic stress, activate the cellular autophagy pathway in the cells, resulting in modified, surplus, or harmful intracellular components being sequestered in autophagosomes and surrendered to the lysosome for the degradation and recycling of intracellular components to regenerate the macromolecule and produce energy during the dynamic program of stem cell proliferation, self-renewal and differentiation [8,15].

Apoptosis is a genetically controlled and evolutionarily conserved form of cell death that is important for normal embryonic development and for the maintenance of tissue homeostasis in the adult organism. Programmed cell death or apoptosis is a genetically determined cell routine in which cells undergo an unexpected decline in homeostasis and functionality, triggering several intracellular pathways and ultimately cell death. The role of Ca^2+^ as a death trigger has been long proposed, and it has been shown that the entry of excess Ca^2+^ into cells might be the mechanism underlying the tissue pathology [1,6]. Ca^2+^ release from the ER, as well as Ca^2+^ entry that is taken up by the mitochondria, is pivotal in triggering apoptotic signals and is one of the mechanisms through which the overexpression of antiapoptotic proteins (or ablation of proapoptotic ones) counteracts cell death. The amount of releasable Ca^2+^—rather than the Ca^2+^ concentration of the ER—seems to be the relevant parameter for the transduction of the death signal, as it controls the ‘amplitude’ of the signal reaching the mitochondria [1,5,8,16,17,18].

## 3. Calcium as Regulator for Apoptosis

Apoptosis is a well-characterized mechanism of programmed cell death and is functionally distinct from autophagy. Notably, in many cell types and disease conditions, the activation of autophagy inhibits the apoptotic mediated cell death, whereas autophagy inhibition activates the apoptotic process [19,20]. However, in some physiological conditions, the proteins originally involved in regulating autophagy might also induce apoptosis [19]. Interestingly, many stimuli that ultimately cause apoptosis also trigger autophagy in the same cell, where autophagy occurs first, and if not resolved, it may proceed to apoptosis [21,22,23]. Apoptosis can be triggered by various cellular signals, especially the intracellular Ca^2+^. Several studies have provided strong experimental evidence that supports the involvement of intracellular Ca^2+^ homeostasis in the induction of apoptosis [24,25,26]. Fleckenstein et al. provided the first evidence that showed the intracellular Ca^2+^ overload caused by excessive Ca^2+^ influx induces cell death in myocytes [27]. Similarly, the lethal effect of intracellular Ca^2+^ overload in cell survival was scrutinized using Ca^2+^ ionophore/chelator in thymocytes. Notably, Ca^2+^ ionophore A23187 mimics the apoptotic effect of glucocorticoid, and the removal of extracellular Ca^2+^ by EGTA diminishes the cytolytic action of glucocorticoid and A23187, suggesting that Ca^2+^ influx and intracellular Ca^2+^ concentration play a crucial role in cell survival/apoptosis [28]. Several other studies have shown that exposing the cells to different apoptotic inducers causes a sustained increase in intracellular Ca^2+^ concentration, which in turn activates the endonuclease and initiates the programed cell death cascade [29,30,31]. The increase in intracellular Ca^2+^ concentration in cells is mainly achieved either by increased Ca^2+^ influx through the plasma membrane Ca^2+^ channels or Ca^2+^ release from the internal storage such as the endoplasmic reticulum (ER) and mitochondria. In addition to intracellular Ca^2+^ overload, other studies have shown that apoptosis can be triggered by both Ca^2+^ depletion from the ER and Ca^2+^ entry through the plasma membrane [32].

## 4. Intracellular Ca^2+^ Stores in Regulating Apoptotic Function

The ER is a major Ca^2+^ storage organelle in mammalian cells, which maintain a concentration of around 1mM compared with 100 nM in the cytosol and mitochondria. Ca^2+^ homeostasis in the ER is essential for the activity of molecular chaperones and the folding of enzymes. Dysregulation in ER Ca^2+^ content affects protein folding, leading to the accumulation of misfolded proteins in the ER and inducing ER stress. Cells activate defense mechanisms such as the unfolded protein response (UPR) to remove the unfolded and/or incorrectly folded accumulated proteins in the ER [33,34]. Therefore, mild or transient ER stress activates several pathways to combat the ER stress, including the activation of chaperones, folding enzymes, reactive oxygen species (ROS) scavengers, and degradative pathways, such as autophagy. In contrast, these pathways can also induce apoptosis when the cells experience severe or prolonged ER stress [35,36].

ER Ca^2+^ homeostasis is tightly regulated by active Ca^2+^ uptake via Ca^2+^ pumps from the cytosol and Ca^2+^ release from the ER by Ca^2+^ leak channels. The Ca^2+^ release from the ER occurs primarily via inositol 1,4,5-trisphosphate (IP_3_) receptor (IP_3_R) or ryanodine receptors (RyRs); however, other Ca^2+^ leak channels also exist in the ER, and these channels are restricted to excitable cell types [37,38]. IP_3_R is predominantly localized on the ER, is one of the major ER Ca^2+^ leak channels, and IP_3_R-mediated Ca^2+^ release induces the release of several pro-apoptotic factors from the mitochondria and activates apoptosis. IP_3_R1-deficient T cells showed resistant to apoptosis induced by dexamethasone, T cell receptor (TCR) stimulation, ionizing radiation, and Fas stimulation [39]. Interestingly, increasing the intracellular Ca^2+^ concentration by thapsigargin makes the IP_3_R1-deficient T cells susceptible to TCR-mediated apoptosis. Interestingly, the removal of external Ca^2+^ has no effect on the TCR-mediated apoptosis in thapsigargin-treated cells, suggesting that ER store depletion is sufficient and Ca^2+^ entry is not required to trigger TCR-mediated apoptosis [39]. In another study, the expression of IP_3_R was upregulated in lymphocytes exposed with apoptotic inducers, and the augmented receptor population was localized to the plasma membrane. Notably, the antisense-mediated silencing of type 3 IP_3_R (IP_3_R3), but not type 1 IP_3_R, inhibited dexamethasone-induced apoptosis in the mouse S49 T-lymphoma cell line, suggesting that Ca^2+^ release via IP_3_R3 is required to trigger autophagy that inhibited apoptosis [29].

Ca^2+^ release from the ER transmits Ca^2+^ to the surrounding mitochondria via the mitochondria-associated ER membranes (MAMs). The stimulation of IP_3_R releases Ca^2+^ from the ER, which generates high-Ca^2+^ microdomains at the ER–mitochondrial junction, leading to the activation of mitochondrial Ca^2+^ uniporter and subsequent mitochondrial Ca^2+^ uptake. Ca^2+^ homeostasis in the mitochondria is required in regulating the activity of Krebs cycle enzymes and transporters, impacting its bioenergetic and biosynthetic function [40,41,42]. Various apoptotic stimuli cause the release of excessive Ca^2+^ from the ER and subsequent Ca^2+^ overload into the mitochondria, resulting in the constant opening of the mitochondrial permeability transition pore (PTP), that leads to mitochondrial swelling with perturbation or rupture of the outer mitochondrial membrane (OMM) and alters the mitochondrial membrane potential [43,44,45]. The rupture of OMM activates the release of several pro-apoptotic proteins such as cytochrome c, apoptosis-inducing factor (AIF), procaspase 9, Smac/DIABLO, and endonuclease G into the cytosol, where it partners with other proteins and initiates the signaling cascade, leading to apoptosis [35,46,47,48,49]. The critical role of Ca^2+^ and apoptosis was further reinforced while studying the mechanism of action of the anti-apoptotic proteins, such as B cell lymphoma 2 (Bcl-2), B cell lymphoma extra-large (Bcl-XL) and myeloid cell leukemia 1 (Mcl-1). Bcl-2 is a critical regulator in controlling apoptotic cell death by neutralizing proapoptotic Bcl-2 family members at the mitochondria. Bcl2 also regulates Ca^2+^ homeostasis in the ER and mitochondria and controls the cell death pathway [50]. Interestingly, Bcl-2 overexpression has been shown to inhibit cell death by decreasing the steady-state Ca^2+^ levels in the ER, thereby reducing Ca^2+^ transfer from the ER to the mitochondria [51]. On the other hand, proapoptotic BAX promotes cell death by increasing the Ca^2+^ mobilization from the ER to the mitochondria. Moreover, the loss of BAX reduced resting ER Ca^2+^ levels that resulted in diminished mitochondrial Ca^2+^ uptake, and were highly resistant to apoptosis [52].

## 5. Calcium Channels/Receptors and Its Role in Autophagy and Apoptosis


### 5.1. Inositol Trisphosphate Receptors

Regarding the regulation of autophagy, the inositol trisphosphate receptor (IP_3_R) represents one of the most studied calcium-permeable channels [53]. The pathway from cAMP to the inositol trisphosphate receptor (IP_3_R) regulates cytosolic Ca^2+^ levels, as IP_3_ binds to IP_3_R on the endoplasmic reticulum (ER), thereby facilitating Ca^2+^ release from the ER [54]. The role of IP_3_R in autophagy is complex, which could activate/inhibit autophagy and has a dual impact [53]. Some studies show that the inhibition of IP_3_R abolished the increase in the autophagy marker LC3 lipidation and GFP-LC3-puncta formation in starvation-induced autophagy [8]. The autophagy protein Beclin1 promotes autophagosome formation, when it is not bound with Bcl_2_ [18]. IP_3_R sensitization increased Beclin1 binding to the IP_3_R and a consequent decrease in Bcl_2_–Beclin1 interaction [8,55,56]. Together, these findings indicate that induced autophagy may require Ca^2+^ release from IP_3_R. In contrast, IP_3_R knockdown or IP_3_R-deficient cells demonstrated higher basal autophagy levels [57,58]. Importantly, the expression of functional IP_3_R3, but not Ca^2+^-impermeable mutant IP_3_R3^D2550A^, was able to rescue elevated autophagy in these cells [57]. These findings suggested the mechanism in which constitutive IP_3_R-mediated Ca^2+^ release is taken up by mitochondria, and this uptake is fundamentally required to maintain mitochondrial bioenergetics and ATP production in resting cells, thereby suppressing autophagy [57]. Another group showed that the IP_3_R pathway negatively regulates autophagy by interacting with the Beclin1/Bcl_2_ complex and/or controlling mitochondrial Ca^2+^ levels [16]. IP_3_R knockdown restored basal autophagy levels and autophagy flux in muscular dystrophy X-linked mouse (mdx) muscle fibers [16]. These reports collectively suggest a complex role for IP_3_R, since both stimulatory and inhibitory functions for IP_3_R toward autophagy have been reported [8,9,15,16]. The reasons for different results could be that IP_3_R could regulate autophagy at different stages, and the effects of IP_3_R depends on multiplicity and cross reactivity factors (such as different cells and their growth conditions, Ca^2+^ signaling, associated proteins and others) that could play additional roles in modulating different functions.

### 5.2. Voltage-Gated Calcium Channels


Besides Ca^2+^ release, Ca^2+^ influx also plays a key role in the regulation of cytosolic Ca^2+^ levels. In excitable cells, voltage-gated calcium channels (VDCCs) are the major contributors for maintaining cytosolic Ca^2+^ levels. Most of the existing reports converge in L-type Ca^2+^ channel antagonists (verapamil, loperamide, nimodipine, nitrendipine and amiodarone). Studies showed that L-type Ca^2+^ channel antagonists can induce mTOR-independent autophagy [59]. L-type Ca^2+^ channel agonists increase cytosolic Ca^2+^, which can activate calpains, and activate the α-subunit of heterotrimeric G proteins G*s*α, which increase in cAMP levels and enhance the IP_3_ production [59]. However, the relation between VDCC and autophagy is a subject of controversy. In the Pompe disease mice model, defective autophagy is particularly prominent and the β 1 subunit of the L-type Ca^2+^ channel is upregulated [60]. An L-type Ca^2+^ channel antagonist (verapamil) reversed the mitochondrial abnormalities and significantly decreased the level of Ca^2+^ and made muscle cells free from autophagic buildup, as observed in Pompe disease [60]. Nevertheless, the normalization of Ca^2+^ levels by verapamil did not have any appreciable effect on autophagy [60]. Interestingly, in contrast to L-type channels, T-type Ca^2+^ channel blockers (mibefradil), inhibit constitutive autophagy and the effect of mibefradil is mimicked by the knockdown of the T-type (Ca_v_3.2) Ca^2+^ channel in cardiomyocytes [61].

### 5.3. Transient Receptor Potential Canonical and Orai1 Channels

SOCE represent one of the major calcium-entry pathways in non-excitable cells. The depletion of Ca^2+^ from the ER causes the ER calcium sensor STIM1 (stromal interaction molecule 1) to translocate to the plasma membrane, where it binds to Orai1/TRPC1, and activating Ca^2+^ entry, which provides Ca^2+^ for ER store refilling as well as for signaling purposes. The possible roles of SOCE/Orai1/TRPC1 in autophagy were inconsistent, with some studies suggesting an inhibitory role for Orai1 in autophagy. The expression of Orai1 decreases in MIR519-induced autophagy. The overexpression of Orai1 decreases the MIR519-mediated induced LC3-II levels carcinoma cells [62]. Another group showed that the knockdown of Orai1 enhances the 5-fluorouracil (5-FU)-induced inhibition of the phosphoinositide 3-kinase-AKT-mTOR pathway and potentiated 5-FU activated autophagic cell death in hepatocarcinoma cells [63]. In contrast, with these studies, other studies have suggested that STIM1/Orai1 stimulates autophagy [64,65,66]. In cardiomyocyte, the silencing of Orai1 could markedly attenuate the Ang II-induced LC3-II accumulation and inhibited autophagy both in vitro and in vivo [64]. It seems that the autophagy-related role of Orai1 and SOCE can be either stimulating or inhibiting depending on the cell type or the stimulation it receives.

TRPC1 and Orai1 are important candidates of SOCE channels that are activated upon store depletion. TRPC1 plays a vital role in hypoxia and nutrient-depletion-dependent autophagy in both excitable and non-excitable cells [67]. The silencing of TRPC1 channels attenuates the increase in autophagy markers and thereby upregulates the apoptotic markers [67,68]. Drugs (tunicamycin and neurotoxins) attenuated autophagy and increased apoptosis through a decrease in TRPC1 function, leading to ER stress [68,69]. In contrast, the restoration of TRPC1 expression increased Ca^2+^ entry and promoted autophagy [68,69]. Together, these findings demonstrated that an increase in intracellular Ca^2+^ via TRPC1 regulates autophagy, thereby preventing cell death. However, besides TRPC1/Orai1, the role of the TRPC3 channel in autophagy remains controversial. Some studies showed that the depletion of TRPC3 lead to an increase in the rate of activation of autophagy in response to supramaximal receptor (CCK8) activation and upon the addition of bile acids. This regulation of autophagy is related to reduced SOCE in *Trpc3*^−/−^ cells [70]. However, studies from our lab showed that silencing of TRPC3 did not affect the hypoxia-induced autophagy, and has no role in modulating SOCE [67]. Similarly, TRPC6 was also shown to modulate autophagy, suggesting that diverse TRPC channels could be involved in modulating different functions (Table 1).

### 5.4. Transient Receptor Potential Melastatin Channels

The TRPM (transient receptor potential melastatin) family is a large subfamily of the TRP superfamily. Recent studies showed the TRPM7 channel played a role in the regulation of basal autophagy (Table 1). When TRPM7 channel expression was induced in a nutrient-rich condition, LC3-II levels increased, indicating the increased level of basal autophagy [71]. In contrast, the silencing/inhibition of the TRPM7 channel in neuronal cells resulted in decreased basal autophagy [71]. In addition, the inhibitory effect of channel inhibitor on basal autophagy was reversed by increasing extracellular Ca^2+^concentration, suggesting that Ca^2+^influx through the TRPM7 channel directly links to basal autophagy. Besides Ca^2+^, TRPM7 is also an important Mg^2+^ channel, and another study indicated that the silencing of the TRPM7 channel increases beclin1 expression and activates autophagy in human bone marrow mesenchymal stem cells (MSCs). Moreover, Mg^2+^ deficiency mimics the effects of silencing TRPM7, and this is in association with the trigger of autophagy without any detectable increase in free Ca^2+^, which suggests Mg^2+^ also contributes to autophagy regulation [72]. The TRPM2 ion channel has been shown to be involved in oxidative-stress-mediated induced autophagy [17,73,74,75]. TRPM2-mediated Ca^2+^ regulates the interplay between ROS and autophagy [74]. Oxidative stress triggers TRPM2-dependent Ca^2+^ influx to inhibit the induction of early autophagy, which renders cells more susceptible to death. TRPM2 also activates the calmodulin-dependent protein kinase II (CaMKII) to phosphorylate Ser295 on Beclin1 [73,74]. The Ser295 phosphorylation of Beclin1 in turn decreased the association between Beclin1 and PIK3C3/VPS34, but induced binding between Beclin1 and BCL_2_ [75]. Together, the above studies demonstrated that oxidative stress activates the TRPM2-Ca^2+^-CAMK2 cascade to phosphorylate Beclin1, resulting in the inhibition of autophagy.

### 5.5. Transient Receptor Potential Vanilloid Channels

The TRPV (transient receptor potential vanilloid) family is a large subfamily of the TRP superfamily. All TRPVs are highly Ca^2+^ selective. TRPV1 has been proposed to regulate autophagy in thymocytes [76,77] (Table 1). The specific TRPV1 agonist CPS can induce autophagy [78], which is through Atg4C pathways and requires a Ca^2+^ increase and ROS generation that induces Atg4C protein oxidation, resulting in AMPK activation [76,78]. The genetic deletion of TRPV1 results in increased basal levels of p62 and in a defective autophagic response to the mTOR inhibitor and autophagy inducer rapamycin [77]. The degradation of TRPV1 in HeLa cells is mediated by autophagy, and this pathway can be amended by cortisol [79]. Moreover, TRPV1-dependent autophagy occurs as a consequence of proteasome inhibition and ER stress triggered by TRPV1. It reduces 20 and 26S proteasome activities and increases p27 expression [77]. Similarly, the activation of TRPV2 also enhances autophagy by stimulating the expression of several genes involved in the autophagic process and in the unfolded protein response [80]. The autophagy genes activated by TRPV2 include ULK2, Atg9B, ATG10, Beclin-1, GARABAP, ATG16 L2, PI3CG and RAB24 genes [81]. The TRPV4 channels also promote autophagy by inhibiting AKT via generating Ca^2+^ signals [82]. Silencing TRPV4 increases apoptosis and inhibits autophagy in HSC-T6 cells. Additionally, the over-expression of TRPV4 induced the activation of autophagy [82].

**Table 1 cells-10-02125-t001:** Calcium channels and their modulation in cellular functions.

Autophagy/Apoptosis	TRP Channels	Used Materials	Molecular Pathway	Reference
Decreased TRPC1 expression subsequently attenuated autophagy along with increased apoptosis.	TRPC1	Neuronal Cells	ER stress	[67]
Depletion of TRPC3 causes activation of autophagy in response to supramaximal CCK8 and to bile acids.	TRPC3	In mice single pancreatic acinar cells	Intracellular trypsin activation and excessive actin depolymerization	[70]
Cannabidiol stimulates autophagy signal transduction via crosstalk between the ERK1/2 and AKT kinases.	TRPV1	Human neuroblastoma SH-SY5Y and murine astrocyte cell lines.	ERK1/2 and AKT kinases	[82,83]
TRPV1 induces autophagy in nitrogen mustard (NM)-caused cutaneous injury.	TRPV1	The HaCaT cells	Ca^2+^/calmodulin-dependent kinase β (CaMKβ), AMP-activated protein kinase (AMPK), unc-51-like kinase 1 (ULK1) pathway	[77]
TRPV1 activation mitigates hypoxic injury in mouse cardiomyocytes by inducing autophagy.	TRPV1	Primary cardiomyocytes isolated from C57 mice	AMPK signaling pathway	[84]
TRPV2 agonist leads to the activation of autophagy.	TRPV2	Glioblastoma stem cell (GSC)	Stimulating the expression of several genes involved in the autophagic process and in the unfolded protein response	[80,81]
TRPV4 channels promote autophagy.	TRPV4	Rat Hepatic stellate cell (HSC)	Inhibiting AKT via generating Ca^2+^ signals	[85]
Oxoglaucine protects against cartilage damage by blocking the TRPV5/CAMK-II/calmodulin pathway to inhibit Ca^2+^ influx and activate autophagy.	TRPV5	In vitro and in vivo- rat model of osteoarthritis	CAMK-II and calmodulin	[86]
Trichostatin A suppresses cervical cancer cell proliferation and induces apoptosis and autophagy through regulation of the PRMT5/STC1/TRPV6/JNK axis.	TRPV6	HeLa and Caski cervical cancer cell lines	JNK pathway	[87]
TRPM2 ion channel has been involved in oxidative stress-mediated induced autophagy.	TRPM2	TRPM2 KO mice	Via oxidative stress and stimulates NADPH oxidase	[74,75,88,89]
TRPM7 channel played a role in regulation of basal autophagy.	TRPM7	Neuronal cells	Ca(2+)/calmodulin-dependent protein kinase kinase β and AMP-activated protein kinase pathway	[71,90]
Silencing TRPM7 trigger autophagy without any detectable increase in free Ca^2+^.	TRPM7	hMSC cells isolated from adult human bone marrow	Via Mg^2+^ channel regulation	[72]

## 6. Autophagy/Apoptosis and Stem Cell Function

Stem cells are specialized and undifferentiated cells with two unique features, including unlimited self-renewable capability and differentiation into all types of cells. Thus, understanding the factors, including the surrounding cellular environment, a signaling system in the cell such as Ca^2+^ signaling and autophagy, can be extended to make them a better candidate for transplantation and organ development and their use in regenerative therapy [34]. The principal tissue reservoir of mammalian adult stem cells is the bone marrow. However, other sources are skin, placenta, adipose tissue and umbilical cord, where these stem cells, especially MSCs, proliferate and differentiate and heal the spoiled tissue during injury [91]. Self-renewal and differentiation requirements drive stem cells’ capacity that is dependent on their ability to have adequate autophagic processes [92]. Impaired autophagy due to a lack of autophagy components such as Atg5 and beclin1 in embryoid bodies (EBs) exhibits a defect in embryoid cavitation and exhibits a deficit in apoptotic corpse engulfment, which is also dependent on calcium. Another study implies that the lack of apoptosis in Bax/Bak double-knockout (DKO) mice also lacks embryonic development [93]. Furthermore, Bax and Bak regulate the type 1 inositol trisphosphate receptor and calcium leak from the endoplasmic reticulum that modulates autophagy [94]. Similarly, beclin 1 also plays a significant role in autophagy and apoptosis pathways. IP_3_R antagonist xestospongin B induces autophagy by disrupting a molecular complex formed by the IP_3_R and Beclin 1, an interaction that is increased or inhibited by the overexpression or knockdown of Bcl-2, respectively [95].

Autophagy is also essential during cartilage differentiation, which also requires calcium, of human umbilical cord mesenchymal stem cells (hUCMSCs) [83]. Research showed that GSK3 positively regulated the mammalian target of rapamycin complex 1 (mTORC1) to suppress autophagic activity, whereas the GSK3β inhibitor prevented the progression of LC3-II/LC3-I. Furthermore, rapamycin, a target for mTOR and an autophagy activator, decreased mTOR expression and improved the chondrogenesis of SMSCs by inducing autophagy. Stem cells cultured in low calcium, however, increased their maintenance [96], which could be due to increased autophagy. Interestingly, MSCs stemness is also supported by mitochondrial fission that is modulated by calcium [97]. Autophagy, particularly mitophagy, transferred impaired or surplus mitochondria to lysosomal degradation, resulting in controlling mitochondrial numbers and mitochondrial quality.

## 7. Autophagy and Apoptosis in Immune Cells

Autophagy is an essential feature for lymphocyte survival and function, and as most of the immune cells require TRPCs/Orai1 channels for their function, they might be crucial for the maintenance of the cellular and metabolic homeostasis of lymphocytes [98]. Several studies have also shown that autophagy regulates the differentiation of lymphocytes by regulating the mitochondria, which is also dependent on calcium, and preventing cell death by controlling the endoplasmic reticulum (ER) function [99]. Autophagy is known to be regulated by the mTOR pathway, along with several autophagy-related proteins that, together with pattern recognition receptors, inflammasome formation, antigen presentation, and LC3-associated phagocytosis, coordinate pathogen clearance [100]. Normally, the autophagy pathway connects mitochondrial and ER content with immune cell functions, which are calcium reservoirs that change the immune cells on its core, and directly affects macrophage and T cell polarization.

### 7.1. Neutrophils

Neutrophils are the most abundant differentiated cells of the immune system, and they are the first responders to act against invading pathogens or host-derived mediators [101]. Mitroulis et al. (2010) were the first to demonstrate that autophagy happens in human neutrophils; their study showed that the action was phagocytosis-dependent and phagocytosis-independent at the same time. Following this group’s findings, numerous studies have shown that autophagy is a key factor in regulating neutrophil functions, including degranulation, metabolism, and neutrophil extracellular trap (NET) formation [102]. The process of autophagy increases exponentially in response to many cellular stresses, such as starvation, endoplasmic reticulum (ER) stress, oxidative stress, and exposure to certain chemicals, radiation, and hypoxia, which are all dependent on calcium [103,104]. Neutrophils can make use of autophagy as an antimicrobial effector function during the initiation of an innate immune response that can be regulated in the process of a memory response [105] and is known to be dependent on calcium entry. Autophagy can be beneficial by fighting various pathogens and preventing their growth and disease onset, but can also be harmful by inducing potent inflammatory responses, including NET formation on the systemic and tissue level [105].

### 7.2. Monocytes/Macrophages

Macrophages are phagocytic antigen-presenting cells (APCs) that are also dependent on autophagy that prevents monocyte apoptosis to differentiation [106]. There are two main soluble mediators that modulate the autophagic responses: damage-associated molecular patterns (DAMPs) and cytokines are key for its function and are dependent on calcium entry [107]. Extracellular or intracellular pattern recognition receptors, including Toll-like receptors, together with advanced glycosylation end product-specific receptor, purinergic P2RX7 receptors and absent in melanoma 2, receive these signals and activate downstream signaling for calcium activation and the initiation of autophagy. Studies have shown that the E3 ligase TNF receptor-associated factor 6-mediated ubiquitination of Beclin 1 is critical for TLR4-triggered autophagy in macrophages, which releases Beclin 1 from its inhibitor BCL2 [108,109,110].

### 7.3. T Cells

Several studies have shown that T cells, even though in small numbers in the cytoplasm, express active autophagic genes. The accumulation of ER and altered calcium mobilization have also been reported in ATG7-deficient mouse T cells [111]. Moreover, the spleens and lymph nodes of mice lacking other autophagic genes also showed a reduction in T cell activation and development with an increased level of apoptosis [112,113]. Autophagy-deficient T cells appear to be unable to regulate organelle quality control; correspondingly, these cells showed a rise in mitochondrial calcium load associated with enhanced levels of reactive oxygen species and cell death.

### 7.4. B Cells

B cells, which generate humoral immunity and immunological memory receive information that is crucial to their physiology and function through cytosolic Ca^2+^ signals. A lack of active autophagic genes causes a decrease in the number of peripheral B cells, suggesting that it is critical for its survival. A study showed that in the absence of autophagy there is an increase in endoplasmic reticulum stress, which results in an increased B lymphocyte-induced maturation protein-1 expression [114]. The role of autophagy in naïve B cell activation is not completely elucidated. Chen and collaborators demonstrated that autophagy is essential for the initiation of an antiviral immune response; however, another group showed that autophagy activity is at the highest in the germinal center B cells, which can suggest that it may have an impact on certain activated stages during B cell responses [115].

### 7.5. Dendritic Cells

It has been shown that autophagy is involved in several dendritic cell (DC) functions, such as maturation, triggers of DC maturation such as Toll-like receptor (TLR) stimulation, antigen presentation, cytokine production, migration and T cell activation, which are also dependent on calcium entry [116]. Baghdadi and his team showed that autophagy reduced antigen presentation and dendritic cell maturation during cancer chemotherapy, while others observed that the inhibition of the autophagic system on regulatory T cells (Tregs) also reduces the maturation of DCs [117,118]. In conclusion, various studies support the belief that autophagy has an inhibitory role in the immunogenic maturation of DCs during maturation. Nevertheless, more investigation needs to be carried out to determine if there is a specific maturation signature under the control of autophagy in DCs.

### 7.6. Nature Killers

Nature killers (NKs) are widely studied for their anti-tumor and anti-inflammatory functions. These cells were initially thought to be large granular lymphocytes with natural cytotoxic ability against tumor cells, but later studies showed them as a separate lymphocyte lineage with both cytotoxicity- and cytokine-producing effector functions [119]. Autophagy was detected in immature NK cells and is essential for the development of these cells by removing damaging mitochondria, resulting in the reduction in the intracellular reactive oxygen species (ROS) levels, which regulate cell death and are dependent on calcium. The lack of properly activated autophagic proteins in NK cells interrupts the initiation of autophagy in immature NKs and impairs the normal development and viral clearance in these cell types, and also is required for bone-marrow-derived mast cell degranulation, but not for differentiation [120,121].

## 8. Autophagy Inhibits Apoptosis in Neuronal Cells

The mammalian cells must recycle their existing molecules and organelles (such as ER, lysosomes that store calcium) and synthesize the new components to combat against various stressors and maintain their lifespan. The process involving the renovation of cellular components in different organs is not well understood, but it is widely recognized that autophagy and apoptosis pathways are critical to maintaining homeostasis and responding to stress stimuli in neurons. Since most of the neurons in the brain are born during embryogenesis, it is unavoidable and essential for the cells to modify and recycle their intracellular components to adopt the different physiological stimuli. Moreover, in post-mitotic cells, neurons are particularly susceptible to the damage caused by the accumulation of abnormal or misfolded proteins because of their inability to clear the protein aggregates by cell division that also requires calcium [122,123]. Several studies have shown that autophagy mediates the degradation of most of the protein aggregates associated with neurodegenerative diseases, including mutant huntingtin (mHTT), α-synuclein (α-syn), tau and TAR DNA-binding protein 43 (TDP43) and defects in the autophagy machinery causes the accumulation of the misfolded protein in the brain [124,125].

Autophagy is a normal physiological process in neurons, and it is constitutively active at the basal level and blocking of basal autophagy causes neuronal death in animal models [126]. Neurons have a highly efficient lysosomal degradation system, which removes autophagosomes rapidly [127]. The critical role of autophagy in neuronal survival was first established using autophagy-specific gene knockout animal models [128,129]. The systemic deletion of ATG5 led to the accumulation of cytoplasmic inclusion bodies in neurons and progressive defects in motor function, and mice died within a day after birth [130]. Similarly, ATG7 knockout causes the accumulation of polyubiquitinated protein-containing inclusion bodies in neurons, although no significant changes in the proteasome function, which is dependent on calcium entry was observed [131]. Notably, the observed inclusion bodies accumulation and vulnerability to cell death in autophagy-deficient neurons vary significantly among different neuronal types, indicating that cell-type-specific mechanisms respond to degrading autophagy substrate and neurodegeneration [131]. Moreover, mutations identified in proteins involving autophagy machinery and calcium genes have been linked to the progression of neurodegenerative diseases in humans [132,133]. Although autophagy is required to maintain the homeostasis in neurons, excessive autophagy can cause brain injury in certain conditions such as perinatal cerebral ischemia or hypoxia–ischemia; thus, the inhibition of autophagy is a potential therapeutic strategy to treat cerebral ischemia [134].

## 9. Role of Autophagy/Apoptosis in Neurodegenerative Diseases

The critical role of autophagy machinery in the pathogenesis of neurodegenerative diseases have been extensively studied over the last few years. Ageing is the primary risk factor for most neurodegenerative diseases; changes in the protein turnover or protein misfolding in neurons will have cumulative effects that manifest later in life [135]. Notably, postmortem brain samples affected by Alzheimer’s disease (AD), Parkinson’s disease (PD), Huntington’s disease (HD) and amyotrophic lateral sclerosis (ALS) indicate that intracellular protein inclusions or extracellular protein aggregates are the common features of neurodegenerative diseases [136,137]. The autophagy pathway is a complex mechanism that includes multiple steps and modes of regulation in neurons for protein degradation, and the impairment of the steps in this network can lead to the accumulation of protein aggregates and cell death.

### 9.1. Autophagy/Apoptosis in Alzheimer’s Disease (AD)

AD is the most common neurodegenerative diseases, characterized by the intracellular accumulation of neurofibrillary tangles containing hyperphosphorylated tau, and the extracellular pathogenic accumulation of β-amyloid (Aβ) plaques [138]. The involvement of autophagy in AD pathogenesis was first reported in 2005 using neocortex biopsy specimens from an AD brain and control brain [138]. Notably, autophagic vacuoles (AVs) were rarely seen in control brain samples and abundant in AD neurites. Moreover, AVs were most frequently observed on the perikarya of affected neurons, particularly those with neurofibrillary tangles [138]. The observed increase in AVs in the AD brain suggested that there may be an impairment in AV biogenesis or transport and their maturation to lysosomes, which could be due to alteration in calcium signaling. Several studies showed that the intracellular accumulation of Aβ significantly decreased upon autophagy activation and increased when inhibiting the autophagy pathway in experimental AD models [139]. The observed changes in the expression of several autophagy-related proteins in brains from AD patients and mouse models also indicate the involvement of autophagy in AD pathology [140]. In this respect, the decreased expression of Beclin 1, a critical protein that regulates the formation of autophagosomes, was observed in the postmortem AD brains and in the APPs-wePS1dE9 transgenic mouse model of AD, which may explain the reduced autophagosome formation correlated with AD [141,142]. Beclin deficiency causes a reduction in neuronal autophagy, increased intraneuronal Aβ accumulation and extracellular Aβ deposition, and subsequent neurodegeneration in APP transgenic mice. Phosphatidylinositol-binding clathrin assembly protein (PICALM/CALM), which is also regulated by calcium, plays a crucial role in clathrin-mediated endocytosis machinery and effects autophagy by regulating the endocytosis of SNARE complexes [142,143].

Presenilin 1 (PS1) is the catalytic subunit of γ-secretase that regulates APP cleavage into Aβ and modulates ER calcium levels [144]. Accumulating evidence suggested that PS1 is required for the autophagy process, and the deletion of PS1 severely impaired the autophagy function, as well as leading to a calcium overload. A disease-causing mutation in PS1 impaired the clearance of autophagy and might account for protein aggregation in AD. PS1 deletion leads to a loss of lysosomal acidification and consequently a loss of lysosomal proteolytic capacity and stability. Autophagosome clearance and substrate proteolysis were severely diminished in PS1-null blastocysts, neurons from mice hypomorphic for PS1 or conditionally depleted of PS1 due to impairment in autolysosome acidification and cathepsin activation. Furthermore, a similar lysosomal/autophagy phenotype was observed in fibroblast from patients with familial AD caused by PS1 mutation, indicating that defects in autophagy machinery may contribute, at least partly, to pathogenic protein accumulation and neuronal death in FAD [144]. The tau protein is predominantly localized in axons, involving axonal microtubule stabilization. Tau is abnormally hyperphosphorylated in the AD brain, which is also due to increased calcium activation and accumulates into intracellular tangles [145]. The ubiquitin–proteasome system (UPS) and autophagy are known to be involved in the degradation of soluble monomeric tau; however, autophagy is the primary mechanism for clearing the oligomeric and insoluble tau aggregates in neurons, aiming to prevent neurodegeneration.

### 9.2. Autophagy/Apoptosis in Parkinson’s Disease (PD)

PD is the second most common neurodegenerative movement disorder characterized by the selective loss of dopaminergic neurons in the substantia nigra pars compacta (SNpc) and the presence of α-synuclein inclusions called Lewy bodies [146]. The dysfunctional lysosomes and accumulation of intracellular autophagosomes were observed in neurons of post-mortem brain samples from PD patients and in vivo PD models, indicating the involvement of autophagy in neurodegeneration [147]. Accumulating evidence indicates that defective autophagy is central to both the etiology and pathogenesis of Parkinson’s disease, and most of the genes involved in developing familial forms of PD are directly associated with the autophagy–lysosomal pathways. Interestingly, both an increase and decrease in calcium levels are observed in PD, but the loss of calcium entry through SOCE channels has been shown to prevent autophagy, whereas the increase in calcium via the voltage gated channels was critical for apoptosis [148,149]. The level of α-synuclein is also critical and autophagy plays a vital role in degrading the α-synuclein in dopaminergic neurons, indicating that defects in autophagic machinery can contribute to the development of synuclein inclusions. The overexpression of α-synuclein affects autophagosome synthesis through the inhibition of GTPase Rab1 [147]. In this context, the observed increase in α-synuclein levels in PD patients is due to triplication of the gene or polymorphisms in the promoter region that can inhibit autophagy and induce neuronal loss.

Interestingly, both wild-type and mutant A53T α-synuclein have been shown to impair autophagy by interacting with high mobility group box 1 (HMGB1) [150,151]. Autophagic stimuli activate the translocation of nucleus HMGB1 to the cytosol, where it interacts with beclin 1, resulting in the dissociation of beclin 1-bcl2 and the subsequent induction of autophagy. Mutations in the acid β-glucocerebrosidase (GBA1) gene encode lysosomal enzyme β-glucocerebrosidase (GCase) are the most common genetic risk factors causing PD. The enhanced aggregation of α-synuclein and impairment in the autophagosome biogenesis and autophagosome-lysosome fusion were observed in the dopaminergic neurons derived from the iPSC of PD patients carrying GBA1 mutations, suggesting an increased level of α-synuclein may cause the autophagic/lysosomal system dysfunction. In addition, the loss of TRPC1 channel led to a decrease in cytosolic calcium levels and showed increased α-synuclein aggregation and inhibition of autophagy [69,148,149]. In addition, pathogenic mutations in ATP13A2 that lead to reduced intracellular calcium levels cause early-onset parkinsonism [152]. Notably, a severe lysosomal alteration was observed in ATP13A2 PD patient-derived fibroblasts that may induce stress to the cell, which eventually activates cell death.

Mutations represent the other most common genetic risk factors in familial forms of PD in the leucine-rich repeat kinase 2 (LRRK2/PARK8) gene [152]. Several studies have shown that LRRK2 involves regulating autophagic machinery as well being involved in calcium homeostasis. Interestingly, the age-dependent degeneration of the nigrostriatal pathway was observed in transgenic mice expression disease-causing G2019S mutant LRRK2 [153,154]. Importantly, this mutant of LRRK2 also altered the ability of the neurons to buffer intracellular calcium levels. Homozygous or compound heterozygous mutations in the PTEN-induced kinase 1 (PINK1) and PRKN (Parkin) have also been associated with early-onset PD. PINK1 and PRKN are functionally connected and involved in a mitochondrial quality control system through a mitophagy-dependent process. *PINK1* deficiency causes the mitochondrial accumulation of calcium*,* resulting in mitochondrial calcium overload that could prevent autophagy and induce apoptosis [155]. The activation of PINK1–PRKN recruits autophagosome membranes via Rab protein and LC3 proteins and is subsequently degraded by the autophagy–lysosomal pathway [156]. Disease-causing mutations in PINK1 or PRKN impair the activation of the PINK1–PRKN-dependent mitophagy pathway, resulting in the accumulation of damaged mitochondria in the cytosol, activating apoptotic mediated cell death, and thereby contributing to neurodegeneration [157].

### 9.3. Autophagy/Apoptosis in Huntington’s Disease (HD)

HD is a neurodegenerative disorder caused by a CAG trinucleotide repeat expansion in the huntingtin (HTT) gene, resulting in the formation and aggregation of an abnormally long polyglutamine, leading to motor dysfunction, behavioral disturbances and cognitive dysfunction [158]. There are many lines of evidence connecting the autophagy and calcium signaling and mutant htt clearance in HD. Mutant htt protein is shown to be accumulated in the cytoplasm and formed cytoplasmic vacuoles. Importantly, autophagy activation decreases the aggregated and soluble monomeric htt and prevent cell loss [159]. Calpains that are members of the family of calcium-activated cysteine proteases also inhibit autophagy, and strategies that reduce calpain activity are shown to increase autophagy and decrease levels of mutant Htt [160]. It has also been shown that wild-type htt serves as a scaffold protein for various types of selective autophagy by recruiting other autophagy proteins such as P61 and ULK1, whereas the mutant htt impaired the autophagic process. Furthermore, htt and its adapter protein huntingtin-associated protein-1 (HAP1) are colocalized in the autophagosomes in neurons, controlling autophagosome dynamics, regulating the molecular transport motors dynein and kinesin to promote efficient retrograde axonal transport. The pharmacological inhibition of the mammalian target of rapamycin (mTOR) by rapamycin induces the mutant htt clearance and decreases the level of soluble proteins and aggregates and cell death in the in vitro HD model [159]. In addition to mTOR sequestration with mutant htt, it has also been shown that mutant htt recruits beclin 1 and impairs its function. Shibata et al. showed that the intracellular accumulation of mutant htt is highly sensitive to beclin 1 level, and the overexpression of beclin 1 significantly reduced the mutant htt accumulation in HD cell models [161]. Moreover, beclin 1 is recruited to cytoplasmic Htt-*N*-terminal product aggregates in HD mouse brain and the striatal samples of HD patients. The sequestration of beclin 1 in htt inclusion might further affect Beclin 1 function and the autophagic degradation of mutant htt, further exacerbating disease progression. Mutant huntingtin also interacts with and inactivates striatal-specific protein Rhes to prevent the Rhes-induced autophagy activation. Interestingly, the knockdown of endogenous Rhes decreases the function of autophagy, whereas overexpression induces autophagy in PC12 cells. Rhes is required for autophagy induction because it robustly binds with Beclin 1 and decreases the inhibitory interaction of Bcl-2 with Beclin 1 [161].

### 9.4. Autophagy/Apoptosis in Amyotrophic Lateral Sclerosis (ALS)

ALS is a rare neurodegenerative disease caused by the aggregation of ubiquitinylated protein that leads to a selective loss of motor neurons in the motor cortex, the brain stem and the spinal cord. Interestingly, alteration in calcium homeostasis has been suggested to contribute to motor neuron demise in ALS [162], which could be due to its regulation of autophagy and apoptosis. A large number of pathogenic mutations within over 30 genes have been associated with the disease, and recently, it has become apparent that many of these genes encode various autophagy receptors such as sequestosome 1 (SQSTM1/p62), optineurin (OPTN) and Ubiquilin 2 (Ubqln2), which are necessary for the recruitment of cargo into autophagosomes for degradation. On the other hand, the pathogenic mutation associated with TAR DNA-binding protein 43 (TDP-43), superoxide dismutase 1 (SOD1), fused in sarcoma/translocated in sarcoma (FUS), and C9ORF72 causes protein misfolding and the accumulation of misfolded protein aggregates, which contribute to motor neuronal loss in ALS [163,164]. Interestingly, the observed accumulation of autophagosomes and impaired proteasome activity in neurons of the spinal cord and brain in ALS patients could explain why the neurons failed to remove the accumulated intracellular aggregates. Emerging evidence also suggests that p62 plays a crucial role in autophagy and selective protein degradation. Interestingly, p62 interacts with mutant SOD1 and facilitates autophagy-mediated mutant SOD1 degradation, whereas mutation in p62 causes the impaired autophagic clearance of mutant SOD1 [165]. The p62 protein interacts with LC3 through its LC3-interacting region (LIR) to facilitate the autophagy-mediated degradation of ubiquitinylated protein aggregates. Similarly, several disease-causing mutations were identified in OPTN. OPTN is an autophagic receptor that regulates autophagosome maturation, the autophagy-mediated degradation of protein aggregates [165]. Mutations in TBK1 have also been associated with both sporadic and familial ALS. The expression of ALS-associated TBK1 mutant in cells blocks autophagosome formation and impairs the clearance of damaged mitochondria. In addition, ALS-causing mutations in OPTN, Q398X, and E478G disrupt the binding of OPTN with myosin VI and affect the autophagosome–lysosome fusion, resulting in autophagosome accumulation [166]. The hexanucleotide repeat expansion in the non-coding region of C9ORF72 is the most common cause of ALS that regulates endosomal trafficking and is colocalized with Rab1, Rab5, Rab7 and Rab11, indicating the involvement of C9ORF72 in autophagosome formation [167].

## 10. Conclusions

Clearly, intracellular Ca^2+^ plays a crucial role in both basal and induced autophagy and in the regulation of apoptosis in both non-excitable and excitable cells. Autophagy is a key component in critical catabolic functions in all cell types. In the innate immune system, the autophagic pathway directly affects cell differentiation and modifies cell functions such as antigen presentation, phagocytosis and cytokine production. Thus, autophagy is responsible for antigen presentation and the production of antimicrobial peptides, as auto-phagolysosomes directly digest pathogens and protect against bacterial/viral infections. Stem cells that are critical for the development of multicellular organisms also depend on autophagy for their function. Similarly, autophagy plays a critical role in preventing protein aggregation and organelle damage that leads to neurodegenerative diseases. Neuronal diseases such as AD, PD, ALS and HD that occur due to impairment in protein trafficking and degradation could be due to the loss of autophagic machinery as discussed here, and thus inducing autophagy could be beneficial for these patients. Calcium is a central regulator that not only modulates autophagy, but could also regulate apoptosis, and thus serves as key modulator for cellular process. Understanding the function of various calcium channels will be critical in understanding their role in these biological processes as well as their dysfunction in various diseases.

## Figures and Tables

**Figure 1 cells-10-02125-f001:**
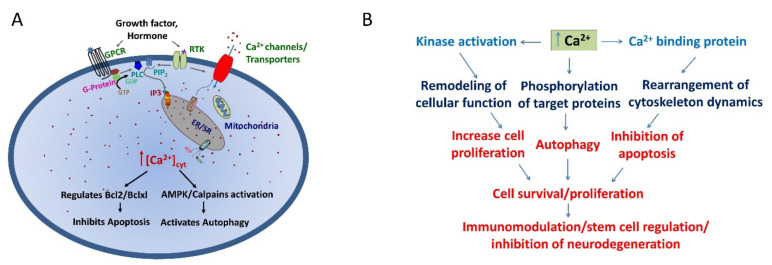
Ca^2+^ signaling and its function in cell survival. Model showing the common Ca^2+^ signaling pathways is shown in (**A**). Cells have several mechanisms for regulating cytosolic Ca^2+^ concentration. The activation of G coupled protein receptor (GPCR or receptor tyrosine kinase (RTK) complex activates PLC, which catalyzes the dissociation of PIP2 to form diacylglycerol and IP_3_. IP_3_ binds to its receptors in the ER (IP_3_R), resulting in the release of the stored Ca^2+^ from the ER. Emptying of the ER Ca^2+^ activates the STIM protein to translocate to the PM and binds to Ca^2+^ channels (TRPCs/Orais). Activation of these Ca^2+^ channels or regulation of other Ca^2+^ channels (TRPs and voltage gated channels) increases cytosolic calcium levels that maintains autophagy and inhibits apoptosis. (**B**) Summary of the pathways that are activated upon increase in cytosolic calcium levels is shown in the schematic diagram.

## Data Availability

Not applicable.

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
