# Peer review of "Calcium Signaling Regulates Autophagy and Apoptosis"

_cells, 2021, doi:10.3390/cells10082125_

Round 1
Reviewer 1 Report
The authors reviewed involvement of calcium channels such as TRP cation, SOCE, and voltage gated calcium channels on autophagy and apoptosis in the neurological diseases. For the aim, the authors summarized physiological functions and families of calcium channels in human. Then, the authors reviewed the action of the calcium channels on autophagy and apoptosis in human. The study is potentially interesting for the journal, and it has also high citation potential. Before decision on the manuscript, the author should give appropriate answer to major comments at the below.
MAJOR REMARKS
- The authors should a table on apoptosis/autophagy and TRP channels. In the table, they should summarize present literature, used materials (cells, animal and human samples), possible molecular pathway (such as caspase activations, Bcl-2, and Bax), and reference.
- In the review paper, the authors reviewed autophagy instead of ‘apoptosis’, because there’re is limited information on apoptosis in the most of sections of the MS. Hence, the authors should add a section namely ‘Calcium channels/receptors and its role in apoptosis’ to the MS.
- In line 222 (Autophagy and function in stem cells). The section out of topic of the MS. The authors should either explain interaction between calcium channels and autophagy in stem cells or they should delete the section.
- In abstract. The authors should extensively express involvement of Ca2+ influx-mediated apoptosis/autophagy in neurodegenerative diseases, because the topic was extensively reviewed in the MS.
- In lines 858 and 868 (Conclusions). The section is too short. Please extend the section and express the future perspectives.
Minor remarks
In title. - In the review the authors focused on SOCE channels. Please revise the title as ‘Calcium channels as regulators for autophagy and apoptosis: Focus on SOCE channels’.
- In line 17, Abstract. Please use ‘, including apoptosis’ instead of ‘such as apoptosis’
- In line 165. The sentence is not clear ‘TRPC1 is an important candidate of SOCE channels as well as Orai1’.
- In line 166. ‘Studies from 165 our lab have..’. There is a citation for the sentence, although it has plural mean.
- In line 461. intracellular reactive oxygen species ROS levels. Please use ‘ROS levels’ instead of ‘intracellular reactive oxygen species ROS levels’.
- In lines 581 (Autophagy in Alzheimer’s disease (AD):..) and 653. Why the sentences are blue?
Reviewer 2 Report
The authors want to review calcium channels as regulators for autophagy and apoptosis, but the structure of the review is not consistent with the topic. More than half of the content is about autophagy, and little else. Therefore, the most important thing to do is to restructure and reorganize the contents of the review.
In addition, a large part of the references cited in the review are not new enough. It is better to cite more references in the last 5 years.
Round 2
Reviewer 2 Report
The biggest problem with this paper is that much of it has nothing to do with calcium signaling, and too much unnecessary content. Content in page 8-19 should be delete or rewtite to stick to the topic of the review.
Author Response
We again thank the reviewer for the careful review of our manuscript. We also agree that in the previous version we have not explicitly indicated the role of calcium in these cells. Thus, as suggested the content on pages 8-19 are rewritten completely to focus on the topic of the review.